# SymMaP: Improving Computational Efficiency in Linear Solvers through Symbolic Preconditioning

## Abstract

Matrix preconditioning is a crucial modern technique for accelerating the solving of linear systems. Its effectiveness heavily depends on the choice of preconditioning parameters. Traditional methods often depend on domain expertise to define a set of fixed constants for specific scenarios. However, the characteristics of each problem instance also affect the selection of optimal parameters, while fixed constants do not account for specific instance characteristics and may lead to performance loss. In this paper, we propose **Sym**bolic **Ma**trix **P**reconditioning (**SymMaP**), a novel framework based on Recurrent Neural Networks (RNNs) for automatically generating symbolic expressions to compute efficient preconditioning parameters. Our method begins with a grid search to identify optimal parameters according to task-specific performance metrics. SymMaP then performs a risk-seeking search over the high-dimensional discrete space of symbolic expressions, using the best-found expression as the evaluation criterion. The resulting symbolic expressions are seamlessly integrated into modern linear system solvers to improve computational efficiency. Experimental results demonstrate that SymMaP consistently outperforms traditional algorithms across various benchmarks. The learned symbolic expressions can be easily embedded into existing specialized solvers with negligible computational overhead. Furthermore, the high interpretability of these concise mathematical expressions facilitates deeper understanding and further optimization of matrix preconditioning strategies.

## 1 Introduction

Linear system problems are widely used in machine learning, physics, engineering, and other scientific disciplines (Leon et al., 2006; LeVeque, 2007). These problems often cannot be solved through analytical or closed-form solutions, rendering the development of efficient numerical algorithms (Demmel, 1997). Among these techniques, matrix preconditioning is one of the most popular approaches to enhance the efficiency of solving linear systems (Trefethen & Bau, 2022; Chen, 2005). During the preconditioning process, the selection of preconditioning parameters—such as the over-relaxation factor in Successive Over-Relaxation (SOR) (Golub & Van Loan, 2013)—plays a crucial role.

In traditional numerical computation, fixed constants for preconditioning parameters are often tailored for specific scenarios based on domain expertise (Chen, 2005). However, apart from problem-specific scenarios, the characteristics of the problem also influence the selection of optimal preconditioning parameters. Choosing a fixed constant as the parameter (Benzi, 2002), on the other hand, would overlook the inherent characteristics of the problem, thereby leading to suboptimal performance. Existing heuristic formulas for predicting parameters, such as Young's formula (Golub & Van Loan, 2013), impose many restrictions on both the input and the problem structure and thus are often not suitable for more generalized scenarios. The above all highlight the urgent need for a more flexible approach that not only accommodates a wider range of scenarios but also dynamically adjusts to the evolving characteristics of the problems.

A good algorithm for discovering the optimal preconditioning parameters must overcome several significant challenges: (C1) Strong Generalization Capability. Different application scenarios and

preconditioning goals demand that the algorithm possesses robust generalization capabilities to adapt effectively across various contexts (Golub & Van Loan, 2013).

(C2) Computational Efficiency. Given the frequent invocation of linear system solvers in scientific computing, any parameter prediction algorithm must incur minimal computational overhead and seamlessly integrate with existing specialized solver libraries (Chen, 2005).

(C3) Algorithmic Transparency. The often opaque nature of prediction algorithms hinders researchers' ability to understand and trust their outcomes, making it difficult to validate and refine the models effectively. Conversely, transparent algorithms allow practitioners to validate and verify the underlying mechanisms, which helps ensure that the prediction results are derived in a logically coherent and justifiable manner (Lipton, 2016).

To address these challenges, we propose **Sym**bolic **Ma**trix **P**reconditioning (**SymMaP**), a novel Recurrent Neural Network (RNN)-based symbolic discovery framework that searches for symbolic expressions of efficient preconditioning parameters. We begin by applying grid search to identify the optimal preconditioning parameters based on task-specific performance metrics. Next, the framework conducts a risk-seeking search within the high-dimensional discrete space of symbolic expressions, with the risk-seeking strategy evaluating the best-found symbolic expression. Finally, these symbolic expressions are integrated into modern solvers for linear systems to enhance their computational efficiency.

The experimental results highlight several advantages of our algorithm:

- **Superior Performance**: Our novel matrix preconditioning approach, based on symbolic regression, consistently outperforms traditional algorithms across various benchmarks.

- **Robust Generalization Capability**: It is widely applicable to predicting diverse matrix preconditioning parameters and optimizing for various objectives.

- **Easy Deployment**: The learned symbolic expressions can be easily embedded into existing specialized linear system solvers with negligible additional computational time, which is friendly to the pure CPU environments.

- **High Interpretability**: The learned strategies are concise, one-line mathematical expressions, facilitating further understanding and optimization of matrix preconditioning algorithms by researchers.

## 2 PRELIMINARIES

### 2.1 MATRIX PRECONDITIONING TECHNIQUE

Matrix preconditioning is a technique employed to accelerate the convergence of iterative solvers and enhance the stability of algorithms. It is generally employed in solving linear systems (Chen, 2005; Golub & Van Loan, 2013). These systems are typically expressed in the form:

$$\boldsymbol{A}\boldsymbol{x} = \boldsymbol{b}. \tag{1}$$

The fundamental idea of preconditioning is to transform the original problem into an equivalent one with better numerical properties. The objectives are twofold: 1. to accelerate the convergence of iterations by altering the spectral distribution of the matrix $\boldsymbol{A}$. 2. to reduce the condition number of the matrix $\boldsymbol{A}$, thereby lessening its ill-conditioning and enhancing the stability of iteration. Specifically, this technique involves finding a preconditioner $\boldsymbol{M}$ that approximates either the inverse of $\boldsymbol{A}$ or some form conducive to iterative solutions (Chen, 2005). Consequently, the original equation 1 is transformed into

$$\boldsymbol{M}\boldsymbol{A}\boldsymbol{x} = \boldsymbol{M}\boldsymbol{b}. \tag{2}$$

The preconditioned matrix $\boldsymbol{M}\boldsymbol{A}$ should have a lower condition number than the original matrix $\boldsymbol{A}$, allowing iterative methods such as Generalized Minimal Residual Method (GMRES) (Saad & Schultz, 1986) to converge more rapidly. Some common preconditioning techniques include the Jacobi, Gauss-Seidel, SOR (Young, 1954) , Algebraic Multigrid (AMG) (Trottenberg et al., 2000), etc.

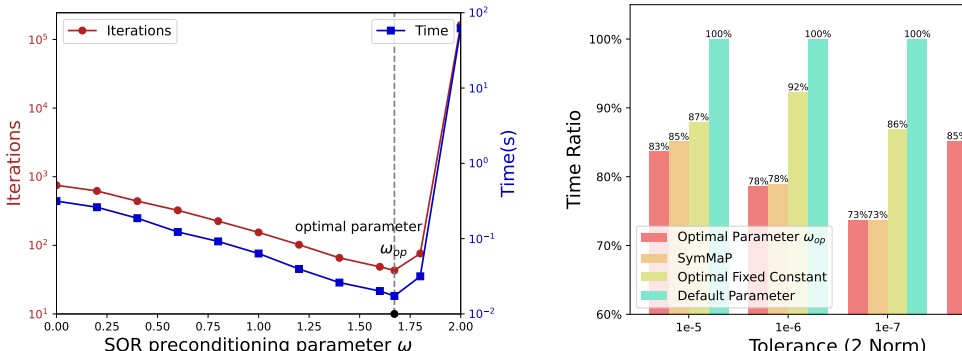

Figure 1: **Left**: Variation in iteration counts and computation times under different SOR preconditioning parameters applied to a linear system from a second-order elliptic PDE. **Right**: Ratio of average computation times at various tolerances to default parameter times under different SOR parameter selection schemes, evaluated on a second-order elliptic PDE dataset.

## 2.2 Prefix Notation and Genetic Programming

Prefix notation is a mathematical format where every operator precedes its operands, eliminating the need for parentheses required in conventional infix notation and simplifying symbolic manipulation. This representation is particularly advantageous in symbolic regression, as it allows mathematical expressions to be expressed as sequences of tokens that can be easily processed by neural networks.

In this notation, operators can be unary (e.g., $\sin$, $\cos$) or binary (e.g., $+$, $-$, $\times$, $\div$), while operands can be constants or variables (Landajuela et al., 2021). Each prefix expression uniquely corresponds to a symbolic tree structure, facilitating the conversion back to the original mathematical expression (Lample & Charton, 2019).

Genetic Programming (GP) is a well-known algorithm that builds on this notation by evolving mathematical expressions that model relationships within data. As an evolutionary computation technique, GP generates solutions without requiring prior knowledge of their structure (Espejo et al., 2009). The expressions are represented as tree structures, with internal nodes corresponding to operators and leaf nodes representing variables or constants (Banzhaf et al., 1998). This representation aligns well with prefix notation, allowing for effective manipulation and optimization.

GP begins with an initial population of randomly generated expression trees and iteratively applies genetic operations—selection, crossover, and mutation—to improve their fitness. Selection identifies the fittest expressions based on a predefined metric (Eiben & Smith, 2015). Crossover combines parts of two parent expressions, while mutation introduces random changes, facilitating the exploration of new potential solutions. Through these iterative processes, GP refines the population of expressions, converging towards optimal solutions that accurately capture the underlying relationships in the data (Espejo et al., 2009; Langdon & Poli, 2013).

## 3 Motivation

The selection of matrix preconditioning parameters plays a crucial role in determining the effectiveness of the preconditioning process (Chen, 2005). To design appropriate algorithmic prediction parameters, we first analyze the optimization space for preconditioning parameter selection and identify the optimal parameters. Subsequently, we examine the unique challenges present in this scenario. Finally, in response to these challenges, we have opted for symbolic learning to guide the parameter selection process.

### 3.1 Motivation for Optimizing Preconditioning Parameters

As illustrated figure 1 (left), the choice of relaxation factors $\omega$ significantly impacts the iteration count and computation time, when solving a second-order elliptic Partial Differential Equation

(PDE) (Evans, 2022) with SOR preconditioning (Young, 1954), There exists an optimal parameter $\omega_{op}$ that minimizes the iteration count and computation time, with specific details available in the Appendix B.2.

To further analyze the optimization space of preconditioning parameters, we evaluated the impact of various parameter selection strategies on preconditioning performance. As shown in Figure 1 (right), the 'Optimal Parameter $\omega_{op}$' represents the parameter that minimizes computation time in each experiment, serving as the theoretical upper limit of our optimization. The 'Optimal Fixed Constant' refers to a fixed constant that minimizes average computation time, and 'Default Parameter' corresponds to the default setting of $\omega = 1$ in the Portable Extensible Toolkit for Scientific Computation (PETSc) (Balay et al., 2024). The gap between the optimal fixed constant and the optimal parameter highlights significant potential for optimizing preconditioning parameter selection, motivating this paper. The performance of our SymMaP algorithm approaches the optimal parameter, demonstrating its accuracy in learning the optimal parameter expression.

## 3.2 CHALLENGES IN PREDICTING EFFICIENT PRECONDITIONING PARAMETERS

The choice of preconditioning parameters directly influences the efficiency of solving linear systems. We aim to develop a universal framework for predicting efficient parameters. However, the context of solving linear systems imposes specific challenges on algorithms that predict preconditioning parameters:

**(C1) Strong Generalization Capability**: Real-world scientific computing scenarios vary significantly. For instance, the grid format and feature selection of PDEs in different physical environments can lead to significant variations in matrix structure (Johnson, 2009), resulting in distinct optimal parameters. Moreover, preconditioning faces various tasks such as reducing computational costs and lowering condition numbers (Chen, 2005). This necessitates that parameter prediction algorithms possess robust generalization capabilities: they should take problem scenarios and characteristics as inputs while being applicable to different preconditioning methods and optimization goals.

**(C2) Computational Efficiency**: Solving linear systems typically relies on Krylov subspace methods implemented in low-level libraries optimized for CPU architectures, such as PETSc (Balay et al., 2024), LAPACK (Anderson et al., 1999). Algorithms like GMRES (Saad & Schultz, 1986) and Conjugate Gradient (CG) (Greenbaum, 1997) iteratively compute the matrix's invariant subspace, favoring single-threaded or limited multi-threaded execution modes. Preconditioning techniques aim to accelerate these solvers without significant additional computational overhead, often adopting implicit iterative formats (e.g., SOR (Chen, 2005)) or utilizing low-cost matrix decompositions (e.g., AMG (Trottenberg et al., 2000)). Therefore, any parameter prediction algorithm must be compatible with CPU environments and seamlessly integrate into existing algorithm libraries while maintaining low computational overhead to preserve the performance gains of preconditioning.

**(C3) Algorithmic Transparency**: Algorithms in scientific computing often require rigorous analysis under mathematical theories. Opaque prediction algorithms could confuse researchers. For instance, the relaxation factor $\omega$ in SOR needs to avoid being too close to 0 or 2 in some scenarios (Agarwal, 2000). This is an issue that opaque algorithms cannot avoid in advance. Moreover, interpretable algorithms can guide researchers to conduct further studies and reveal the underlying mathematical structures of problems. Therefore, these pose challenges to the transparency and interpretability of the parameter prediction algorithms.

## 3.3 SYMBOLIC LEARNING TO PRECONDITIONING PARAMETER SELECTION

Symbolic learning focuses on extracting explicit mathematical expressions from data, enabling the generation of analytical models that link system properties to optimal preconditioning parameters. This approach eliminates manual tuning and enhances solver efficiency by directly predicting parameters based on system characteristics. Introducing symbolic learning into matrix preconditioning addresses the challenges of selecting optimal parameters by offering a generalizable, efficient, and transparent solution.

Firstly, symbolic learning can accommodate various types of input parameters and can specifically tailor symbolic expression learning for different preconditioning methods and optimization goals (Cranmer et al., 2020), thereby meeting the requirement for broad applicability in scientific

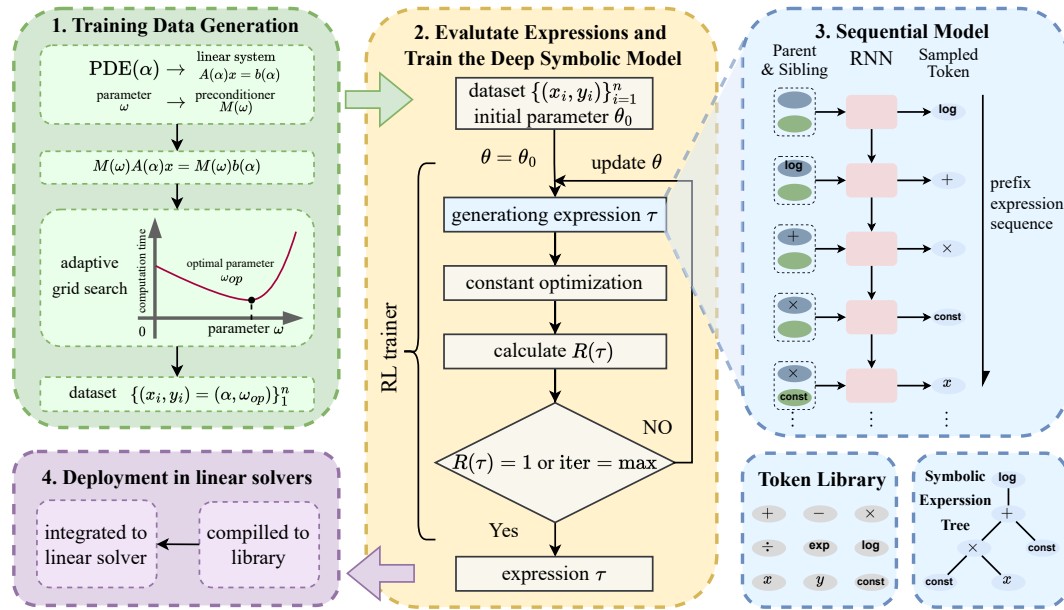

Figure 2: Illustration of how SymMaP discovers efficient symbolic expressions for preconditioning parameters. Part 1 demonstrates the acquisition of optimal parameters and dataset generation; Part 2 illustrates the training process of the RL-based deep symbolic discovery framework; Part 3 shows how the sequential model generates symbolic policies; Part 4 presents the deployment of symbolic expressions.

computing tasks (**Challenge C1**). Secondly, the explicit expressions derived are computationally lightweight and can be quickly evaluated at runtime, integrating seamlessly into existing CPU-based algorithm libraries like PETSc (Balay et al., 2024) with almost no overhead (**Challenge C2**). Thirdly, symbolic learning provides transparent and interpretable models (Rudin, 2019), allowing researchers to understand the influence of parameters within existing theoretical frameworks and identify potential numerical stability issues. This interpretability fosters trust in the algorithm's predictions and supports further theoretical exploration (**Challenge C3**).

# 4 METHOD

In this study, we introduce a novel framework, SymMaP, for symbolic discovery in matrix preconditioning by utilizing RNN. As shown in Figure 2, we first obtain the optimal preconditioning parameters for the given scenario through a grid search to construct a training dataset. Then we employ an RNN to generate symbolic expressions in prefix notation, which are then evaluated for their fitness. The RNN is trained using a reward function based on the performance of the generated expressions. By optimizing the RNN parameters to maximize this reward function, we generate symbolic expressions that approximate the desired mathematical functions. Finally, we deploy the learned symbolic expressions into linear system solvers. The detailed steps are as follows and pseudocode is provided in the Appendix C.

## 4.1 INPUT FEATURES AND TRAINING DATA GENERATION

**Input Features**. We discuss solving parameterized PDEs, a common scenario in linear systems. These problems are generated using methods like uniform random generation, Gaussian Random Fields (GRF), and truncated Chebyshev polynomials (details are provided in Appendix D.1). We use these parameters as input features for symbolic learning in SymMaP.

**Training Data Generation**. For each generated linear system, we determine its optimal preconditioning parameters via adaptive grid search. Taking the SOR preconditioning as an example—with relaxation factors $\omega$ ranging from 0 to 2 and aiming to minimize computation time—we start with

a coarse grid (e.g., step size 0.01) to compute iterations for each factor. We then select the factors with the fewest computation time and perform a finer grid search (e.g., step size 0.001) around them to find the optimal relaxation factor.

By generating a series of linear systems and finding the optimal preconditioning parameters for each, we create the necessary training data. Each data point comprises: 1. the problem generation parameters $x_i$; 2. the optimal preconditioning parameters $y_i$. $i = 1, 2, \ldots, n$ and $n$ is the number of data. Our goal is for symbolic learning to discover an expression mapping the problem parameters to the optimal preconditioning parameters.

## 4.2 The Generation of Symbolic Expressions

**Token Library**. For SymMaP, we define the library of mathematical operators and operands as $\{+, -, \times, \div, \text{sqrt}, \exp, \log, \text{pow}, 1.0\}$. Although other operators such as poly, $\sin$ and $\cos$ are frequently used (Udrescu & Tegmark, 2020), we decided to exclude them because they offer limited explanatory power in matrix preconditioning and significantly increase the time and memory consumption during training.

After converting the mathematical expressions into prefix notation, we leverage this tokenized representation as a pre-order traversal of the expression tree (Zaremba & Sutskever, 2014). In each iteration, the RNN receives a pair consisting of a parent node and a sibling node as inputs and outputs a categorical distribution over all possible next tokens. The parent node refers to the last incomplete operator that requires additional operands to form a complete expression. The sibling node, in the context of a binary operator, represents the operand that has already been processed and incorporated into the expression. In cases where no parent or sibling node is applicable, they are designated as empty nodes. This structured input method enables the RNN to maintain contextual awareness and effectively predict the sequence of tokens that form valid mathematical expressions.

**The Sequential Model**. During the generation of a single symbolic expression, the RNN emits a categorical distribution for each "next token" at each step. This distribution is represented as a vector $\psi_{\boldsymbol{\theta}}^{(i)}$, where $i$ denotes the $i$-th step and $\boldsymbol{\theta}$ represents the parameters of the RNN. The elements of the vector correspond to the probabilities of each token, conditioned on the previously selected tokens in the traversal (Petersen et al., 2019):

$$\psi_{\boldsymbol{\theta}}^{(i)}(\boldsymbol{\tau}_i) = \boldsymbol{p}(\boldsymbol{\tau}_i | \boldsymbol{\tau}_{1:i-1}; \boldsymbol{\theta}). \tag{3}$$

Here, $\boldsymbol{\tau}_i$ denotes the index of the token selected at the $i$-th step. The probability of generating the entire symbolic expression $\boldsymbol{\tau}$ is then the product of the conditional probabilities of all tokens (Petersen et al., 2019; Landajuela et al., 2021):

$$\boldsymbol{p}(\boldsymbol{\tau} | \boldsymbol{\theta}) = \prod_{i=1}^{N} \psi_{\boldsymbol{\theta}}^{(i)}(\boldsymbol{\tau}_i). \tag{4}$$

**Optimization of Constants:** The library $\mathcal{L}$ incorporates a 'constant token,' which allows for the inclusion of various constant placeholders within sampled expressions. These placeholders serve as the parameters $\xi$ in the symbolic expression. We seek to find the optimal values of these parameters by maximizing the reward function: $\xi^* = \arg\max_{\xi} R(\tau; \xi)$, utilizing a nonlinear optimization method. This optimization is executed within each sampled expression as an integral part of computing the reward, prior to each training iteration.

## 4.3 The Reward Function

Once a symbolic expression is fully generated (i.e., the symbolic tree reaches all its leaf nodes), we evaluate its fitness by calculating the Normalized Root-Mean-Square Error (NRMSE), a metric frequently used in genetic programming symbolic discovery (Schmidt & Lipson, 2009). The NRMSE is defined as

$$\text{NRMSE} = \frac{1}{\sigma_y} \sqrt{\frac{1}{n} \sum_{i=1}^{n} (y_i - \hat{y}_i)^2}, \tag{5}$$

where $\hat{y}_i$ is the predicted value for the $i$-th sample, $y_i$ is the optimal preconditioning parameter, $\sigma_y$ is the standard deviation of the target values $y$, and $n$ is the number of data. To bound this fitness value between 0 and 1, we apply a squashing function:

$$R(\boldsymbol{\tau}) = \frac{1}{1 + \text{NRMSE}}. \tag{6}$$

Our objective is to maximize $R(\boldsymbol{\tau})$, thereby minimizing the NRMSE and improving the accuracy of the generated expressions.

### 4.4 THE TRAINING ALGORITHM

Although the objective function is well-defined, it is important to note that $R(\boldsymbol{\tau})$ is not a deterministic value but a random variable dependent on the RNN's parameters $\boldsymbol{\theta}$. Therefore, the key challenge is to establish an appropriate criterion for evaluating this random variable, and then apply gradient-based optimization methods accordingly.

**Risk-seeking Policy.** It is natural to consider the expectation of the reward function, i.e. $\mathbb{E}_{\boldsymbol{\tau} \sim p(\boldsymbol{\tau};\boldsymbol{\theta})}[R(\boldsymbol{\tau})]$, as the objective function to optimize. We can easily obtain

$$\nabla_{\boldsymbol{\theta}} \mathbb{E}_{\boldsymbol{\tau} \sim p(\boldsymbol{\tau};\boldsymbol{\theta})}[R(\boldsymbol{\tau})] = \mathbb{E}_{\boldsymbol{\tau} \sim p(\boldsymbol{\tau};\boldsymbol{\theta})}[R(\boldsymbol{\tau}) \nabla_{\boldsymbol{\theta}} \log \boldsymbol{p}(\boldsymbol{\tau};\boldsymbol{\theta})]. \tag{7}$$

by applying the "log-integral" trick (Williams, 1992). Thus, even though the expectation of the reward function is not directly differentiable with respect to $\boldsymbol{\theta}$, we can approximate the gradient using the sample mean.

In the context of symbolic regression, model performance is often driven by a few exceptional results that outperform others by a significant margin (Petersen et al., 2019; Tamar et al., 2015). With this in mind, we adopt a risk-seeking policy, which aims to maximize:

$$J(\boldsymbol{\theta}, \varepsilon) = \mathbb{E}_{\boldsymbol{\tau} \sim p(\boldsymbol{\tau};\boldsymbol{\theta})}[R(\boldsymbol{\tau}) | R(\boldsymbol{\tau}) > Q(\boldsymbol{\theta}, \varepsilon)]. \tag{8}$$

Here, $Q(\boldsymbol{\theta}, \varepsilon)$ is the $(1 - \varepsilon)$-quantile of the reward distribution under parameter $\boldsymbol{\theta}$, i.e.

$$Q(\boldsymbol{\theta}, \varepsilon) = \inf\{q \in \mathbb{R} | \text{CDF}(R(\boldsymbol{\tau}); \boldsymbol{\theta}) \geq 1 - \varepsilon\}, \tag{9}$$

where $\text{CDF}(R(\boldsymbol{\tau}); \boldsymbol{\theta})$ refers to the cumulative distribution function. From this, the gradient of $J(\boldsymbol{\theta}, \varepsilon)$ can be derived as (Petersen et al., 2019):

$$\nabla_{\boldsymbol{\theta}} J(\boldsymbol{\theta}, \varepsilon) = \mathbb{E}_{\boldsymbol{\tau} \sim p(\boldsymbol{\tau};\boldsymbol{\theta})}[(R(\boldsymbol{\tau}) - Q(\boldsymbol{\theta}, \varepsilon)) \nabla_{\boldsymbol{\theta}} \log \boldsymbol{p}(\boldsymbol{\tau};\boldsymbol{\theta}) | R(\boldsymbol{\tau}) > Q(\boldsymbol{\theta}, \varepsilon)]. \tag{10}$$

This gradient can be estimated using Monte Carlo sampling:

$$\nabla_{\boldsymbol{\theta}} J(\boldsymbol{\theta}, \varepsilon) \approx \hat{g} \triangleq \frac{1}{\varepsilon N} \sum_{i=1}^{N} (R(\boldsymbol{\tau}^{(i)}) - \tilde{Q}(\boldsymbol{\theta}, \varepsilon)) \nabla_{\boldsymbol{\theta}} \log \boldsymbol{p}(\boldsymbol{\tau}^{(i)}; \boldsymbol{\theta}) \cdot 1_{R(\boldsymbol{\tau}^{(i)}) > \tilde{Q}(\boldsymbol{\theta}, \varepsilon)}, \tag{11}$$

$\tilde{Q}(\boldsymbol{\theta}, \varepsilon)$ is the empirical $(1 - \varepsilon)$-quantile of the reward function. By concentrating on the top $\varepsilon$ percentile of samples, SymMaP emphasizes optimizing the best-performing solutions in preconditioning, thereby obtaining the optimal symbolic expressions for preconditioning parameters.

### 4.5 DEPLOYMENT IN LINEAR SOLVER

After the training process, we obtained a symbolic formula for predicting the preconditioning parameters. The learned formula is exceptionally concise and incurs minimal computational cost. Therefore, we directly compile the learned policy into a lightweight shared object using a simple script and then integrate it into the linear system solver package.

## 5 EXPERIMENTS

We conducted comprehensive experiments to evaluate the SymMaP framework, organized into three primary sections: 1. Assessment of three different preconditioners and optimization goals across various datasets to determine the effectiveness of SymMaP algorithm, 2. Analysis of associated

computational overhead and the interpretability of the learned symbolic expressions, 3. Ablation studies of SymMaP.

**Preconditioners**: We considered three different preconditioners and various optimization metrics: 1. SOR preconditioner with the relaxation factor $\omega$ (Golub & Van Loan, 2013); 2. SSOR preconditioner with the relaxation factor $\omega$ (Golub & Van Loan, 2013); 3. AMG preconditioner with the threshold parameters $\theta_T$ (Trottenberg et al., 2000).

**Datasets**: We examined linear systems arising from three distinct classes of PDEs: 1. Darcy flow problems consist of symmetric matrices, generated by the finite difference method (Li et al., 2020; Kovachki et al., 2021); 2. Second-order elliptic PDEs also consist of symmetric matrices, generated by the finite difference method (Evans, 2022; Bers et al., 1964); 3. Biharmonic equations consist of non-symmetric matrices, generated by the finite element method. And non-symmetric matrices are not amenable to SSOR and AMG preconditioning (Ciarlet & Raviart, 1974; Barrata et al., 2023).

**Baselines**: We compared SymMaP against various parameter selection methods for preconditioning. Specifically, the comparison involved the following scenarios: 1. No matrix preconditioning, 2. Default parameters in PETSc (Balay et al., 2024), 3. Fixed constants, 4. Optimized fixed constants.

**Experiment Settings**: To ensure consistent evaluations, all preconditioning was implemented in the C version of the PETSc library (Balay et al., 2024). Experiments were executed using the GMRES algorithm (Saad & Schultz, 1986) within a standardized computing environment.

Details on preconditioners, the mathematical forms of datasets, and the runtime environment are available in Appendices B, D.1, and D.2, respectively. Information on the generation of training datasets for the following experiments and parameters of the SymMaP algorithm are outlined in Appendices D.3 and D.4. For an introduction to related work, see Appendix A.

## 5.1 MAIN EXPERIMENTS

In these experiments, we optimized relaxation factors $\omega$ in both SOR and SSOR preconditioning, and threshold parameters $\theta_T$ in AMG preconditioning. For SOR, we identified $\omega$ values that minimize computation time, forming the training dataset for SymMaP to learn symbolic expressions that optimize computational times for solutions. In SSOR, we used a hybrid metric combining normalized computation times and iteration counts to determine optimal $\omega$ values aimed at optimizing both metrics simultaneously. Similarly, for AMG, we selected $\theta_T$ values that minimize the condition number of preconditioned matrices.

Table 1: Comparison of average computation times (seconds) for SOR with different $\omega$ selections, and tolerance is 1e-7. SymMaP 1 and SymMaP 2 are the two learned expressions that achieved the highest reward function scores, with the best-performing method highlighted in bold.

| Dataset | Matrix size | No precondition | PETSc default $\omega = 1$ | Fixed constant $\omega = 0.2$ | Fixed constant $\omega = 1.8$ | Optimal constant | SymMaP 1 | SymMaP 2 |
|---|---|---|---|---|---|---|---|---|
| Biharmonic | $4.2 \times 10^3$ | 7.67 | 2.04 | 4.86 | 1.60 | 1.31 | **1.24** | 1.26 |
| Darcy Flow | $1.0 \times 10^4$ | 33.1 | 13.5 | 17.5 | 9.91 | 9.54 | **8.50** | 8.60 |
| Elliptic PDE | $4.0 \times 10^4$ | 31.3 | 21.0 | 21.4 | 17.5 | 16.6 | **15.8** | 16.3 |

Table 2: Comparison of average computation times and iterations for SSOR with different $\omega$ selections, and tolerance is 1e-7. SymMaP 1 and 2 are the first two expressions learned. Results are displayed as "time (seconds) / iteration", with the best method in bold.

| Dataset | Matrix size | No precondition | PETSc default $\omega = 1$ | Fixed constant $\omega = 0.2$ | Fixed constant $\omega = 1.8$ | Optimal constant | SymMaP 1 | SymMaP 2 |
|---|---|---|---|---|---|---|---|---|
| Darcy Flow | $4.9 \times 10^3$ | 4.18 / 8596 | 0.488 / 1068 | 0.757 / 1685 | 1.09 / 2322 | 0.448 / 960 | **0.412 / 936** | 0.523 / 1226 |
| Elliptic PDE | $4.0 \times 10^4$ | 23.9 / 5281 | 10.5 / 2369 | 14.7 / 3534 | 8.72 / 1926 | 8.68 / 1864 | **7.70 / 1666** | 7.74 / 1714 |

Experimental results indicate that SymMaP consistently outperforms all others across all experimental tasks. For SOR, Table 1 shows that SymMaP reduces computation times by up to 40% compared to PETSc's default settings and by 10% against the optimal constants. In SSOR, Table 2 shows that it cuts computation time and iteration counts by up to 27% and 30%, respectively, over PETSc's

Table 3: Comparison of average condition numbers for preconditioned matrices using different threshold parameter $\theta_T$ selections in AMG. SymMaP 1 and 2 are the first two expressions learned, with the best method in bold.

| Dataset | Matrix size | No precondition | PETSc default $\theta_T = 0$ | Fixed constant $\theta_T = 0.2$ | Fixed constant $\theta_T = 0.8$ | Optimal constant | SymMaP 1 | SymMaP 2 |
|---------|-------------|-----------------|-------------------------------|----------------------------------|----------------------------------|------------------|----------|----------|
| Darcy Flow | $1.0 \times 10^4$ | 752862 | 8204 | 19146 | 11426 | 7184 | **4824** | 5786 |
| Elliptic PDE | $4.0 \times 10^4$ | 6792 | 184.6 | 205.4 | 212.5 | 182.8 | **168.8** | 170.3 |

defaults, and achieves reductions of 11% in time and 10% in iterations compared to optimal constants. For AMG, Table 3 shows that SymMaP lowers the condition number by up to 40% relative to PETSc's defaults and 32% against the optimal constants.

These results highlight SymMaP's ability to effectively derive high-performance symbolic expressions for various preconditioning parameters, showcasing its broad applicability and strong generalization across different preconditioning tasks.

## 5.2 COMPARISON WITH NEURAL NETWORK PERFORMANCE

To analyze the deployment overhead and predictive performance of the SymMaP algorithm, we conducted a comparison with predictions made directly by a Multilayer Perceptron (MLP). Specifically, we implemented an MLP with three fully connected layers, using Mean Squared Error (MSE) as the loss function and ReLU as the activation function. Both symbolic expression and MLP were executed in a CPU environment to simulate a scientific computing context.

Table 4: Comparison of the runtime required for symbolic expression and MLP to predict the SOR relaxation factor and the subsequent average solution time for linear systems, using the Darcy Flow dataset with a matrix size of $10^3$ and tolerance is 1e-5.

|  | Runtime (s) | Solution time (s) |
|--------|-------------|-------------------|
| MLP | 5.1e-5 | 7.1e-1 |
| Symbol | 1.1e-5 | 7.1e-1 |

As shown in Table 4, the runtime of symbolic expressions learned by SymMaP was only 20% of that of the MLP, primarily due to the poor performance of neural networks in a pure CPU environment, highlighting SymMaP's computational efficiency under these conditions. Furthermore, the average solution times for parameters predicted by both symbolic expressions and MLP were closely matched. This demonstrates that symbolic expressions possess equivalent expressive capabilities to neural networks in this scenario, effectively approximating the optimal parameter expressions.

## 5.3 INTERPRETABLE ANALYSIS

Table 5: Symbolic expressions learned from the main experiments

| Precondition | Dataset | Symbolic expression |
|--------------|---------|---------------------|
| SOR | Biharmonic | $1.0 + 1.0/(4.0 + 1.0/x_2) + 1.0/x_1$ |
| SOR | Elliptic PDE | $1.0 + 1.0/(x_2 + 1.0 + 1.0/(x_2 + 4.0))$ |
| SOR | Darcy Flow | $1.0 + 1.0/(x_4 + 1.0)$ |
| SSOR | Elliptic PDE | $1.0 + 1.0/(x_2 + 1.2)$ |
| AMG | Elliptic PDE | $(x_1 x_3 + 1)/7$ |

In Table 5, we report a subset of the learned symbolic expressions, with the mathematical significance of the related symbols detailed in Appendix E.1. These symbolic expressions are notably more concise and selective, not utilizing all candidate parameters and symbols, which aids researchers in analyzing their underlying relationships.

For instance, in the context of SOR and SSOR preconditioning, empirical evidence suggests that smaller relaxation factors should be chosen when diagonal components are relatively small. Our

experimental findings corroborate this: for the second-order elliptical PDE dataset, the symbolic expressions derived for SOR and SSOR preconditioning depend solely on $x_2$, with larger $x_2$ values leading to smaller predicted relaxation factors, exemplified by $1.0 + \frac{1.0}{(x_2+1.2)}$. Here, $x_2$ represents the coupling coefficient of the elliptical PDE, which directly influences the relative size of the non-diagonal components of the generated matrix, whereas other coefficients have minimal impact. As the coupling coefficient increases, the relative numerical of the non-diagonal components increases, and the diagonal components reduce correspondingly, aligning with empirical observations.

Additionally, further examples include using SymMaP to optimize the relaxation factor for SOR iterations. Training data were generated from the symmetric positive definite matrices portion of the second-order elliptic PDE dataset. By inputting the spectral radius $\rho(\boldsymbol{A})$ into SymMaP, it accurately achieved the theoretically optimal relaxation formula, the Young's formula (Young, 1954):

$$\omega_{\text{opt}} = \frac{2}{1 + \sqrt{1 - \rho(\boldsymbol{A})^2}}. \tag{12}$$

Note: The SOR iteration discussed here is a direct iterative method for solving linear systems and not a preconditioning technique; it is distinct from the SOR preconditioning discussed in the main experiments, though both require setting a relaxation factor.

These experimental outcomes demonstrate that SymMaP can derive interpretable and efficient symbolic expressions for parameters, further aiding researchers in understanding and exploring the underlying mathematical principles.

## 5.4 Ablation Experiments

Table 6: Ablation study examining the selection of mathematical operators, comparing the effects on preconditioning and training times. The first column lists the selected operators, the second column shows the condition numbers of preconditioned matrices derived from AMG parameter predictions on the Darcy flow dataset (lower is better), and the third column displays SymMaP training times.

| Functionset | Condition number | Training time(s) |
|---|---|---|
| $+, -, \times, \div, \text{poly}$ | 6803.8 | 15351 |
| $+, -, \times, \div, \text{sqrt}, \exp, \log, \text{pow}, 1.0$ | 7086.9 | 703.17 |
| $+, -, \times, \div, \text{sqrt}, \exp, \log, \sin, \cos, \text{pow}, 1.0$ | 7172.6 | 635.82 |
| $+, -, \div, 1.0, \text{pow}$ | 7241.8 | 703.26 |
| $+, -, \times, \div, \text{sqrt}, \text{pow}, 1.0$ | 7271.1 | 746.80 |
| $+, -, \times, \div, \text{pow}, 1.0$ | 7301.4 | 702.46 |

We conducted an ablation study using SymMaP to evaluate the impact of different mathematical operator selections, as described in Table 6. In the main experiments, We utilized the operator set $\{+, -, \times, \div, \text{sqrt}, \exp, \log, \text{pow}, 1.0\}$ listed in the second row. The results indicate that this selection of operators achieves a balance between predictive performance and training time efficiency, meeting our expectations. Furthermore, experiments detailing the performance of SymMaP in relation to variations in learning rate, batch size, and dataset Size are documented in Appendix E.2.

## 6 Conclusions and Future Work

In this paper, we propose SymMaP, a deep symbolic discovery framework designed for predicting efficient matrix preconditioning parameters. Experiments show that SymMaP can predict high-performance parameters and is applicable across a variety of preconditioning and optimization objectives. Additionally, SymMaP is easy to deploy with virtually no additional computational overhead. Future work will focus on optimizing preconditioning for specific matrix structures, such as symmetric and upper triangular matrices. We also aim to analyze the mathematical significance of the learned symbolic expressions from a theoretical perspective, such as exploring the impact of problem characteristics on the solution process through pseudospectral analysis. Furthermore, plans to extend SymMaP to additional preconditioning methods (e.g., ILU, ICC) are underway. We are confident in the symbolic model's immense potential for broad real-world applications, especially in matrix preconditioning.

## 7 Code of Ethics and Ethics Statement

This paper adheres to the ICLR Code of Ethics. The research focuses on developing a more efficient matrix preconditioning parameter prediction framework. It does not involve human subjects, personal data, or sensitive information that could raise concerns regarding privacy, security, or fairness. Furthermore, no potential conflicts of interest, legal compliance issues, or harmful applications were identified in this study.

## 8 Reproducibility

For the sake of reproducibility, we have included essential codes in the supplementary materials, covering dataset generation, the algorithm's source code, and performance evaluation scripts. However, it's worth noting that the current code version lacks structured organization. Should this paper be accepted, we commit to reorganizing the codes for improved clarity. Additionally, in Appendix C, we provide pseudocode for our algorithm. In Appendix D, we offer a detailed explanation of our experimental setups.

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

## A  RELATED WORK

### A.1  MACHINE LEARNING FOR ALGORITHM DISCOVERY

Machine learning has the potential to uncover implicit rules beyond human intuition from training data, enabling the construction of algorithms that outperform handcrafted programs. Approaches to algorithm discovery in machine learning encompass symbolic discovery, program search, and more. Specifically, program search focuses on optimizing the computational processes of algorithms. For example, Mankowitz et al. (2023) explores the discovery of faster sorting algorithms, while Chen et al. (2024) investigate efficient optimization algorithms.

In contrast, symbolic discovery aims to search within the space of small mathematical expressions rather than computational streams (Petersen et al., 2019; Landajuela et al., 2021). This approach is analogous to an extreme form of model distillation, where knowledge extracted from black-box neural networks is distilled into explicit mathematical expressions. Traditional methods for symbolic discovery have relied on evolutionary algorithms, including genetic programming (Poli et al., 2008). Recently, deep learning has emerged as a powerful tool in this domain, offering enhanced representational capacity and new avenues for solving symbolic discovery problems (Schmidt & Lipson, 2009; Cranmer et al., 2020).

### A.2  NEURAL NETWORKS FOR MATRIX PRECONDITIONING

Recent studies have explored the use of neural networks to improve matrix preconditioning techniques. Greenfeld et al. (2019); Luz et al. (2020); Taghibakhshi et al. (2021) demonstrate the effectiveness of neural networks in refining multigrid preconditioning algorithms, thus streamlining the computational process. Götz & Anzt (2018) utilized Convolutional Neural Networks (CNNs) for the optimization of block Jacobi preconditioning algorithms, while Stanaityte (2020) developed corresponding Incomplete Lower-Upper Decomposition (ILU) preconditioning algorithms leveraging machine learning insights.Although these algorithms achieved impressive results, they still face challenges such as limited interpretability and reduced computational efficiency when deployed in pure CPU environments. This paper attempts to address these issues by incorporating symbolic learning into the framework.

## B  DETAILED INTRODUCTION OF MATRIX PRECONDITIONING

### B.1  OVERVIEW OF MATRIX PRECONDITIONING METHODS

- **Jacobi Method**: The Jacobi preconditioner utilizes only the diagonal elements of a matrix to precondition a linear system. By approximating the inverse of the diagonal matrix, this method is computationally simple and effective for systems with strong diagonal dominance. However, its convergence rate can be slow, and its performance diminishes for poorly conditioned or weakly diagonally dominant matrices. The Jacobi method is typically used as a baseline for comparison with more sophisticated preconditioners (Saad, 2003).

- **Gauss-Seidel (GS) Method**: The Gauss-Seidel preconditioner improves upon the Jacobi method by considering both the lower triangular and diagonal parts of the matrix in a sequential manner. Unlike the Jacobi method, which updates all variables simultaneously, the GS method updates each variable in sequence using the most recent values. This leads to faster convergence, especially for diagonally dominant matrices. However, the GS method can still struggle with poorly conditioned systems, and its forward-only approach can limit performance in some applications (Saad, 2003).

- **Successive Over-Relaxation (SOR)**: The SOR method builds on the Gauss-Seidel method by introducing a relaxation factor $\omega$ to accelerate convergence. This factor allows for over-relaxation ($\omega > 1$) or under-relaxation ($\omega < 1$), tuning the method for faster performance on certain types of problems. SOR can significantly reduce the number of iterations needed for convergence compared to both the Jacobi and GS methods, but choosing the optimal relaxation factor is problem-dependent (Young, 1954).

- **Symmetric Successive Over-Relaxation (SSOR)**: SSOR is a symmetric version of the SOR method, where relaxation is applied in both forward and backward sweeps of the matrix. This bidirectional process improves stability and is well-suited for use with iterative solvers like the conjugate gradient method, which requires symmetric preconditioners. SSOR's symmetry ensures that the preconditioner maintains the properties needed for efficient and stable convergence, making it a popular choice for symmetric positive-definite systems (Golub & Van Loan, 2013).

- **Algebraic Multigrid (AMG)**: AMG is an advanced preconditioning technique designed to handle large, sparse systems of linear equations, especially those arising from the discretization of partial differential equations. Unlike traditional methods, AMG operates on multiple levels of the matrix structure, coarsening the matrix to form a hierarchy of smaller systems that are easier to solve. Solutions on the coarser grids are then interpolated back to the finer grids. This multilevel approach makes AMG highly efficient for large-scale problems, as it can dramatically reduce the number of iterations needed to achieve convergence. AMG is often used in combination with methods like SSOR or Gauss-Seidel as a smoother on each grid level, and it is particularly effective in cases where the problem exhibits a multiscale nature (Ruge & Stüben, 1987).

**Relationship Among Jacobi, GS, and SOR Methods**: The Jacobi method is the simplest of the three, using only diagonal information. The GS method improves upon the Jacobi method by using both diagonal and lower triangular matrix elements to achieve faster convergence. SOR further refines the GS method by introducing a relaxation factor to optimize the update process. Both the GS and SOR methods can be seen as iterative improvements on the Jacobi method, with SOR offering a more flexible and potentially faster alternative by adjusting the relaxation factor. SSOR extends SOR symmetrically, making it suitable for use in more advanced iterative solvers like the conjugate gradient method (Saad, 2003; Golub & Van Loan, 2013).

## B.2 Parameters in Matrix Preconditioning

The choice of preconditioning parameters significantly influences the effectiveness of the preconditioning process, especially in the iterative solving of linear systems (Chen, 2005). Below, we discuss three specific preconditioning techniques—SOR, SSOR, and AMG—focusing particularly on how their key parameters affect the preconditioning results.

### B.2.1 Relaxation Factor $\omega$ in SOR and SSOR Methods

In the SOR preconditioning method, the relaxation factor $\omega$ is a critical parameter that determines the acceleration of iteration. SOR evolves from the Gauss-Seidel method by introducing $\omega$ to speed up convergence. The SOR iteration formula is given by:

$$\boldsymbol{x}^{(k+1)} = (\boldsymbol{D} + \omega \boldsymbol{L})^{-1} \left[ (1 - \omega) \boldsymbol{D} \boldsymbol{x}^{(k)} + \omega \boldsymbol{b} - \omega \boldsymbol{U} \boldsymbol{x}^{(k)} \right], \tag{13}$$

where $\boldsymbol{D}$, $\boldsymbol{L}$, and $\boldsymbol{U}$ are the diagonal, strictly lower triangular, and strictly upper triangular parts of the matrix $\boldsymbol{A}$, respectively (Golub & Van Loan, 2013).

The SSOR preconditioning method can be represented by the following formula:

$$\boldsymbol{M}_{\text{SSOR}} = \frac{1}{\omega(2 - \omega)} (\boldsymbol{D} - \omega \boldsymbol{U}) \boldsymbol{D}^{-1} (\boldsymbol{D} - \omega \boldsymbol{L}), \tag{14}$$

where $\boldsymbol{M}_{\text{SSOR}}$ constitutes the preconditioner, and $\boldsymbol{D}$, $\boldsymbol{L}$, $\boldsymbol{U}$, and $\omega$ are defined similarly to their roles in the SOR method. This symmetrical formulation enhances the stability and effectiveness of the preconditioning, particularly benefiting symmetric positive-definite matrices by optimizing the convergence properties of the iterative solver (Golub & Van Loan, 2013).

The choice of $\omega$ directly impacts the speed of convergence and the condition number of the matrix. Different problems and scenarios often require different choices of $\omega$, which typically need to

be determined based on the specific properties of the problem and through numerical experimentation (Golub & Van Loan, 2013). In the PETSc library, the default relaxation factor $\omega$ for both SOR and SSOR is set to 1, at which point SOR degenerates to GS preconditioning.

### B.2.2 THRESHOLD PARAMETERS $\theta_T$ IN AMG

In the AMG method, the threshold parameter $\theta_T$ determines whether the non-zero elements of the matrix are "strong" enough to be considered in the construction of a coarse grid during the multigrid process. This parameter is crucial for establishing the connectivity between coarse and fine grids in the hierarchical multilevel structure (Ruge & Stüben, 1987).

The AMG method solves the equation system through multiple levels of grids, each corresponding to a coarser version of the original problem. During this process, the threshold parameter is used to determine whether a given non-zero matrix element is strong enough to keep the corresponding grid points connected during coarsening.

- A lower threshold often leads to more elements being considered as strong connections, which might increase the complexity of the coarse grid but can help preserve the essential characteristics of the original problem, thus improving the efficiency and convergence of the multigrid method.

- A higher threshold might result in fewer strong connections, thereby reducing the complexity of the coarse grid. However, this can weaken the effectiveness of the AMG method, especially in maintaining the characteristics of the original problem.

Different values of $\theta_T$ directly influence the condition number of the preconditioned matrix. Selecting the appropriate threshold parameter typically involves considering the specific structure and characteristics of the problem, and adjustments are made through experimental fine-tuning to achieve the optimal balance (Trottenberg et al., 2000). In the PETSc library, the default threshold parameter $\theta_T$ is set to 0.

## C  ALGORITHM PSEUDOCODE

---

**Algorithm 1** RNN-based Symbolic Discovery Process

---

**Input:** RNN with parameter $\boldsymbol{\theta}$, the library of tokens $\mathcal{L}$.
$\boldsymbol{\tau} \leftarrow [\,]$
parent(0), sibling(0) $\leftarrow$ empty node
$x_0 \leftarrow$ parent(0)$||$sibling(0)               $\triangleright$ $x$ is the concatenation of parent and sibling nodes.
$h_0 \leftarrow 0$.                                      $\triangleright$ Initialize hidden state of RNN.
**for** $t = 1, 2, \cdots$ **do**
    $(\psi_t, h_t) \leftarrow$ RNN$(x_{t-1}, h_{t-1}; \boldsymbol{\theta})$.       $\triangleright$ $\psi_t$ is the categorical distribution of the next token.
    $\psi_t \leftarrow$ ApplyConstraint$(\psi_t, \mathcal{L}, \boldsymbol{\tau})$       $\triangleright$ Regularize the distribution.
    Sample token $\boldsymbol{\tau}_t \sim \psi_t$
    **if** Arity$(\boldsymbol{\tau}_t) > 0$ **then**                $\triangleright$ Arity$(\boldsymbol{\tau}_i)$ denotes the number of operands of $\boldsymbol{\tau}_i$.
        parent$(t) \leftarrow \boldsymbol{\tau}_t$
        sibling$(t) \leftarrow$ empty node
    **else**                    $\triangleright$ When Arity$(\boldsymbol{\tau}_t) = 0$, go back to the last incomplete operator node.
        count $\leftarrow 0$
        **for** $i = t, t-1, \ldots, 1$ **do**                     $\triangleright$ Backward iteration.
            count $\leftarrow$ count + Arity$(\tau_i)$ $-1$
            **if** count = 0 **then**
                parent$(t) \leftarrow \boldsymbol{\tau}_i$
                sibling$(t) \leftarrow \boldsymbol{\tau}_{i+1}$
                **break**
            **end if**
        **end for**
        **if** count $= -1$ **then break**                $\triangleright$ The expression sequence is complete.
        **end if**
    **end if**
    $x_t \leftarrow$ parent$(t)||$sibling$(t)$
**end for**
**Output:** Prefix expression sequence $\boldsymbol{\tau}$.

---

**Algorithm 2** Deep Symbolic Optimization for Matrix Preconditioning Parameter

---

**Input:** RNN with initial parameter $\boldsymbol{\theta}_0$, the library of tokens $\mathcal{L}$, batch size $N$, iteration number $J$, risk factor $\varepsilon$, and learning rate $\alpha$.
$\boldsymbol{\theta} \leftarrow \boldsymbol{\theta}_0$
$j \leftarrow 0$
**repeat**
    **for** $i = 1, 2, \ldots, N$ **do**
        $\boldsymbol{\tau}^{(i)} \leftarrow$ SymbolicDiscover$(\boldsymbol{\theta}, \mathcal{L})$
        $\xi^* \leftarrow \arg\max\{\xi \text{ in } \boldsymbol{\tau} \text{ as constant placeholder} : R(\boldsymbol{\tau}; \xi)\}$        $\triangleright$ Constant optimization.
        $\boldsymbol{\tau}^{(i)} \leftarrow$ ReplaceConstant$(\boldsymbol{\tau}^{(i)}, \xi^*)$
        Compute $\hat{g}_1$ using $\boldsymbol{\tau}^{(i)}$ and $\boldsymbol{\theta}$.                  $\triangleright$ See Eq. equation 11.
        Compute $\hat{g}_2$ as entropy gradient.
        $\boldsymbol{\theta} \leftarrow \boldsymbol{\theta} + \alpha(\hat{g}_1 + \hat{g}_2)$                         $\triangleright$ Update the parameter.
        Train model: update $p_{\boldsymbol{\theta}}$ via PPO by optimizing $J(\boldsymbol{\theta}; \epsilon)$.
    **end for**
**until** $j = J$ or *convergence*
**Output:** The best symbolic expression $\boldsymbol{\tau}^*$.

---

# D EXPERIMENT SETTINGS

## D.1 DATASETS

**1. Darcy Flow Problem**

We consider two-dimensional Darcy flows, which can be described by the following equation (Li et al., 2020; Rahman et al., 2022; Kovachki et al., 2021; Lu et al., 2022):

$$-\nabla \cdot (K(x,y)\nabla h(x,y)) = f,$$

where $K$ is the permeability field, $h$ is the pressure, and $f$ is a source term which can be either a constant or a space-dependent function.

In our experiment, $K(x,y)$ is generated using truncated Chebyshev polynomials. We convert the darcy flow problem into a system of linear equations using the central difference scheme of Finite Difference Methods (FDM) (LeVeque, 2007). The coefficients of the Chebyshev polynomials serve as input features for our symbolic learning algorithm.

**2. Second-order Elliptic Partial Differential Equation**

We consider general two-dimensional second-order elliptic partial differential equations, which are frequently described by the following generic form (Evans, 2022; Bers et al., 1964):

$$\mathcal{L}u \equiv a_{11}u_{xx} + a_{12}u_{xy} + a_{22}u_{yy} + a_1 u_x + a_2 u_y + a_0 u = f,$$

where $a_0, a_1, a_2, a_{11}, a_{12}, a_{22}$ are constants, and $f$ represents the source term, depending on $x, y$. The variables $u, u_x, u_y$ are the dependent variable and its partial derivatives. The equation is classified as elliptic if $4a_{11}a_{22} > a_{12}^2$.

In our experiments, $a_{11}, a_{22}, a_1, a_2, a_0$ are uniformly sampled within the range $(-1, 1)$, while the coupling term $a_{12}$ is sampled within $(-0.01, 0.01)$. We then select equations that satisfy the elliptic condition to form our dataset. Similar to the approach with the darcy flow problem, we convert the PDE into a system of linear equations using the central difference scheme of FDM. The coefficients $a_0, a_1, a_2, a_{11}, a_{12}, a_{22}$ serve as input features for our symbolic learning algorithm.

**3. Biharmonic Equation**

We consider the biharmonic equation, a fourth-order elliptic equation, defined on a domain $\Omega \subset \mathbb{R}^2$. The equation is expressed as follows (Ciarlet & Raviart, 1974; Glowinski & Pironneau, 1979; Barrata et al., 2023):

$$\nabla^4 u = f \quad \text{in } \Omega,$$

where $\nabla^4 \equiv \nabla^2\nabla^2$ represents the biharmonic operator and $f = 4.0\pi^4 \sin(\pi x)\sin(\pi y)$ is the prescribed source term.

In our experiments, we construct the dataset by varying the solution domain $\Omega$. We utilize the discontinuous Galerkin finite element method from the FEniCS library to transform this problem into a system of linear equations (Barrata et al., 2023). The parameters of the domain serve as input features for our symbolic learning algorithm.

## D.2 ENVIRONMENT

To ensure consistency in our evaluations, all comparative experiments were conducted under uniform computing environments. Specifically, the environments used are detailed as follows:

1. Environment (Env1):
   - Platform: Windows11 version 22631.4169, WSL
   - Operating System: Ubuntu 22.04.3
   - CPU Processor: AMD Ryzen 9 5900HX with Radeon Graphics CPU, clocked at 3.30GHz
2. Environment (Env2):
   - Platform & Operating System: Ubuntu 18.04.4 LTS
   - CPU Processor: Intel(R) Xeon(R) Gold 6246R CPU at 3.40GHz

- GPU Processor: GeForce RTX 3090 24GB
- Library: CUDA Version 11.3

Speed tests for solving linear systems were performed in Env 1, while all training related to symbolic learning was conducted in Env 2.

### D.3 TRAINING DATA GENERATION

We employed an adaptive grid search to generate the training dataset. Initially, we traversed a coarse grid, sampling every 0.05, and from this dataset, we selected the three points with the smallest values. Subsequently, we conducted a finer grid search around these points, sampling every 0.001, to identify the point with the minimum value, which we designated as our optimal parameter. Particularly, after experimental validation confirmed the dataset's convexity, we utilized a binary search sampling method for a dataset derived from the second-order elliptic equation's SOR preconditioning. Starting with points at 0.0, 1.0, and 2.0, we compared these values. If the value at 0.0 was lowest, we computed at 0.5; if at 2.0, then at 1.5; and if at 1.0, then at both 0.5 and 1.5. This process was repeated until achieving a minimum point with a precision of 0.001.

For SOR preconditioning, we evaluated second-order elliptic equations, Darcy flow equations, and biharmonic equations, with solution time as the metric for optimal preprocessing parameters, achieved by minimizing solution time using the previously described grid method. In SSOR preconditioning, applied to second-order elliptic and Darcy flow equations, we utilized a hybrid metric that combined normalized computation time and iteration counts, aiming to simultaneously optimize both iteration counts and solution times. For AMG preconditioning, also examined with second-order elliptic and Darcy flow equations, we used the condition number of the preconditioned matrix as the metric, where a lower value indicates better performance.

### D.4 PARAMETERS OF THE SYMMAP

**Experimental Setup**. SymMAP is implemented using the LSTM architecture with one layer and 32 units. More details about the hyperparameters are provided in Table 7.

Table 7: Hyperparameters of SymMAP (Default Model)

| Hyperparameter | Value |
|---|---|
| Number of LSTM layers | 1 |
| Number of LSTM units | 32 |
| Number of training samples | 2,000,000 |
| Batch size | 1,000 |
| Risk factor $\varepsilon$ | 0.05 |
| Minimal expression length | 4 |
| Maximal expression length | 64 |
| Learning rate | 0.0005 |
| Weight of entropy regularization | 0.03 |

**Restricting searching space**. We employ specific constraints within our framework to streamline the exploration of expression spaces effectively and ensure they remain within practical and manageable bounds:

1. **Bounds on expression length.** To strike a balance between complexity and manageability, we set boundaries for expression lengths: a minimum of 4 and a maximum of 64 characters. This ensures that expressions are neither overly trivial nor excessively complicated.

2. **Constant combination.** We restrict expressions such that the operands of any binary operator are not both constants. This is out of the simple intuition that, if both operands are constants, the combination of the two can be precomputed and replaced with a single constant.

3. **Inverse operator exclusion.** We preclude unary operators from having their inverses as children to avoid redundant computations and meaningless expressions, such as in $\log(\exp(x))$.

4. **Trigonometric Constraints.** Expressions involving trigonometric operators should not include descendants within their formulation. For instance, $\sin(x + \cos(x))$ is restricted because it combines trigonometric operators in a way that is uncommon in scientific contexts.

# E    SUPPLEMENTARY EXPERIMENTS

## E.1    INTERPRETABLE ANALYSIS DETAILS

Table 8: Symbolic expressions learned from the main experiments

| Precondition | Dataset | Symbolic expression |
|---|---|---|
| SOR | Biharmonic | $1.0 + 1.0/(4.0 + 1.0/x_2) + 1.0/x_1$ |
| SOR | Elliptic PDE | $1.0 + 1.0/(x_2 + 1.0 + 1.0/(x_2 + 4.0))$ |
| SOR | Darcy Flow | $1.0 + 1.0/(x_4 + 1.0)$ |
| SSOR | Elliptic PDE | $1.0 + 1.0/(x_2 + 1.2)$ |
| AMG | Elliptic PDE | $(x_1 x_3 + 1)/7$ |

As shown in Table 8, the variables are defined as follows: in the first row, $x_1$ and $x_2$ represent the size of the boundary for PDE solutions; in the second row, $x_2$ represents the coefficient of a second-order coupling term; in the third row, $x_4$ is the coefficient of the fourth x-term multiplied by the first y-term in a two-dimensional Chebyshev polynomial; in the fourth row, $x_2$ again denotes the coefficient of a second-order coupling term; in the fifth row, $x_1 x_3$ signifies the coefficient of a second-order non-coupling term.

## E.2    ANALYSIS OF HYPERPARAMETERS

The performance of SymMaP is primarily influenced by the learning rate of the RNN, batch size, and dataset size. We conducted experiments to study the impact of these hyperparameters.

**Symbolic Learning RNN Parameters**:

Table 9: Performance comparison of SymMaP under various symbolic learning RNN parameters (lower condition numbers are preferable). The experiment focuses on optimizing AMG preconditioning coefficients in the Darcy Flow dataset.

| Learning Rate | Batch Size | Condition number | Training time(s) |
|---|---|---|---|
| | 500 | 6780 | 1173.09 |
| 0.01 | 1000 | 5168 | 863.51 |
| | 2000 | 6898 | 522.80 |
| | 500 | 5935 | 1104.16 |
| 0.001 | 1000 | 11774 | 676.40 |
| | 2000 | 5935 | 505.85 |
| | 500 | 4718 | 1026.45 |
| 0.0005 | 1000 | 5935 | 703.17 |
| | 2000 | 5935 | 549.36 |
| | 500 | 12228 | 1324.00 |
| 0.0001 | 1000 | 7508 | 837.18 |
| | 2000 | 6884 | 603.62 |

Results in Table 9 indicate that an appropriate combination of RNN learning rate and batch size can enhance performance.

**Dataset size**:

Table 10: Performance comparison of SymMaP across varying dataset sizes (lower condition numbers indicate better performance). The experiment evaluates the optimization of AMG preconditioning coefficients for the Darcy Flow dataset.

| Dataset size | Condition number | Training time (s) |
|---|---|---|
| 10 | 7032 | 669.68 |
| 50 | 6980 | 737.80 |
| 100 | 4892 | 812.02 |
| 500 | 3811 | 699.54 |
| 1000 | 5345 | 703.17 |

Table 10 demonstrates that increasing the dataset size enhances the performance of symbolic expressions learned by SymMaP, as expected.

