# OpenReview forum: "SymMaP: Improving Computational Efficiency in Linear Solvers through Symbolic Preconditioning"
_ICLR.cc/2025/Conference — ICLR 2025 Conference Withdrawn Submission_

### Official Review · Reviewer_57eX · 2024-11-01

**Soundness:** 3
**Presentation:** 3
**Contribution:** 2
**Rating:** 5
**Confidence:** 5

**Summary:**

This paper addresses the problem of matrix preconditioning, which is a crucial ingredient in the iterative solution of linear systems. In particular, the authors focus on the parameters of some preconditioners and propose a machine-learning approach to determining these parameters. They base their approach on parameterized PDEs, such that it will predict preconditioner parameters given PDE parameters. The authors construct a training dataset and use it to learn symbolic regression formulas for these parameters. They argue that symbolic regression is more efficient and interpretable than other regression techniques.

**Strengths:**

- The paper addresses practical problems.

- The symbolic approach has the potential to reveal the relationship between PDE parameters and preconditioner parameters, leading to new mathematical discovery and analysis.

- The symbolic regression performance is competitive with neural network regression.

**Weaknesses:**

- The empirical evaluation of the proposal is conducted on either too simplistic but less effective preconditioners (SOR and SSOR), or a widely used preconditioner (AMG) with only one single parameter. It would be more informative if the authors experimented with more AMG parameters (such as the choice of the smoother and other coarsening parameters) and conducted a sensitivity analysis.

- The generation of training data is costly: (# data points) * (# grid searches) * (cost of one preconditioned solve)

- The training of symbolic regression can also be costly because of the sample efficiency of reinforcement learning.

- It needs to be clarified of the relationship between genetic programming in section 2.2 and the rest of the paper.

**Questions:**

- The authors use the condition number as the evaluation metric for the AMG experiments, which is hard to compute for large matrices. Why not use computation time instead, which is the most straightforward metric? Would the conclusion be different depending on which metric to use? Note that condition number does not necessarily correlate with time, since coarsening generally poses tradeoffs between preconditioning time and convergence speed.

- Continued from the above question: When constructing the training set for AMG, did the authors use time, iteration, or condition number to determine optimal parameters?

- How do the following three times compare: time of generating the training data, time of training the symbolic regression, and time of one preconditioned solve?

- How many data points are needed to train symbolic regression?

- For the interpretability analysis in 5.3, do the expressions come from SymMaP 1 in Tables 1 to 3? Are there trade-offs between the preconditioning performance of a symbolic formula and the interpretability?

---

### Official Review · Reviewer_83ou · 2024-11-04

**Soundness:** 1
**Presentation:** 1
**Contribution:** 1
**Rating:** 3
**Confidence:** 5

**Summary:**

The authors introduce Symbolic Matrix Preconditioning (SymMaP), a method that identifies symbolic expressions for efficient preconditioning parameters. These generated expressions can be seamlessly integrated into modern solvers with minimal computational overhead.

**Strengths:**

The proposed framework involves defining the optimal preconditioning parameters, subsequently searching for symbolic expressions, and integrating these expressions into the modern solver.

**Weaknesses:**

* Although the application of the proposed approach in this context may be novel, the reasoning behind why it could outperform a class of preconditioners based on graph neural networks (https://proceedings.mlr.press/v202/li23e.html, https://arxiv.org/abs/2405.15557) is not evident.
* The authors test their framework using three datasets (Biharmonic, Darcy Flow, and Elliptic PDE) with minimal variation in matrix size. It would be beneficial to validate their approach using the SuiteSparse Matrix Collection (https://sparse.tamu.edu/).
* I found some sections a bit challenging to follow. Could the authors consider reorganizing the paper or providing more detailed explanations for each step of SymMaP to enhance clarity?
* The values in the columns labeled "SymMap 1" and "SymMap 2" in tables 1, 2, 3 are not clear and would benefit from additional information to clarify the results.
* A comparison of the performance in a CPU environment using an MLP is not adequate to make a definitive assertion `symbolic expressions possess equivalent expressive capabilities to neural networks in this scenario, effectively approximating the optimal parameter expressions`.

**Questions:**

See weaknesses.

---

### Official Review · Reviewer_X8gT · 2024-11-09

**Soundness:** 2
**Presentation:** 2
**Contribution:** 2
**Rating:** 3
**Confidence:** 3

**Summary:**

The manuscript presents a novel approach to quasi-optimal preconditioner construction based on symbolic regression. The authors perform extensive numerical simulations to identify the optimal parameters for given linear systems through grid search. The pairs (a parameter for a linear system and the corresponding optimal preconditioner parameter) compose the training dataset. Then, the combination of RL trainer and RNN for symbolic regression fits the analytical expression for the optimal preconditioner parameter. Experimental results demonstrate that the presented pipeline gives such a preconditioner that, on average, the runtime for linear solvers is smaller for different PDE classes.

**Strengths:**

Disclaimer: I am not an expert in symbolic regression, so advances in the study from this perspective could not be well-identified.

The manuscript's main strength is its attempt to apply the general symbolic regression technique to the preconditioner construction problem. The presented pipeline looks non-trivial, although the objective function is standard. In experiments, the presented approach generates preconditioners that establish faster convergence of the linear solver. In addition, the SyMMap could reconstruct the optimal expression for the $\omega$ in SSOR for positive definite matrices.

**Weaknesses:**

The main weakness of this study is that it misses the crucial step of using the derived symbolic expression to generate the optimal preconditioner. I have carefully checked Figure 2 and do not find an explicit connection between the expression $\tau$ and the compiled preconditioner in the library. Some remarks are given in Section 5.3; however, the presented expressions depend on unclear variables $x_1, x_2, x_3$, so how to use them in preconditioner construction is unclear.

In addition, I can list the following weaknesses:
1. The manuscript does not explicitly present the parametrizations of considered PDEs and how these parameters are passed as input to RNN. Also, the authors ignore details on how training and testing sets are prepared.
2. I did not find the name of the linear solver used to evaluate the preconditioners, e.g., CG, GMRES, or smth else.
3. The incomplete Cholesky/LU preconditioner is not included in the comparison, although it is among the most powerful.
4. The authors do not report the runtime for training the presented pipeline (although for Darcy, the runtime is given in Table 6). Moreover, they do not discuss how many linear systems are needed to solve with the generated preconditioner to pay off the training costs compared to classical approaches like SSOR with the optimal parameter or ILU. Tables 1 and 2 show a gain in runtime, which is good, but how much time does the training of symbolic regression require?
5. For unknown reasons, the authors include a comparison with MLP. However, a more interesting comparison is replacing RNN with Transformer architecture and analyzing the results in the performance of linear solver and training runtime.
6. No theoretical guarantees on the performance of such an approach or motivation of the presented pipeline are presented, so the robustness of this approach remains unclear.

**Questions:**

Some questions are in the previous section on the weaknesses of the submission; other questions are given below.

1. What are alternative approaches to constructing optimal preconditioner parameters? Please add a paragraph to place your work in the context of learning preconditioners from data or similar techniques. For example:
- https://proceedings.mlr.press/v202/li23e.html
- https://sc18.supercomputing.org/proceedings/workshops/workshop_files/ws_lasalss102s2-file1.pdf
- https://arxiv.org/abs/2405.15557
- https://arxiv.org/abs/2401.02016
- https://arxiv.org/abs/1806.06045

2. What are the spectrum properties of the preconditioned matrix with the generated preconditioner? It would be interesting to observe whether they only reduce the condition number or additionally increase spectrum clustering. Condition numbers for the preconditioned matrices are presented in Tables 3 and 6, but only for limited types of PDEs.

3. What was the $\epsilon$ parameter used in experiments, and does it significantly affect the training runtime/performance?

---

### Note · Authors · 2024-11-17

I have read and agree with the venue's withdrawal policy on behalf of myself and my co-authors.